# Abdominal volume index, waist-to-height ratio, and waist circumference are optimal predictors of cardiometabolic abnormalities in a sample of Lebanese adults: A cross-sectional study

**Myriam Abboud**[1]*, **Suzan Haidar**[2], **Nadine Mahboub**[2], **Dimitrios Papandreou**[3], **Rana Rizk**[4,5]

1 College of Natural and Health Sciences, Zayed University, Dubai, United Arab Emirates, 2 Department of Nutrition and Food Sciences, Lebanese International University, Beirut, Lebanon, 3 Department of Clinical Nutrition & Dietetics, College of Health Sciences, University of Sharjah, Sharjah, United Arab Emirates, 4 Department of Natural Sciences, Lebanese American University, Byblos, Lebanon, 5 Institut National de Santé Publique, d'Epidémiologie Clinique, et de Toxicologie (INSPECT-LB), Beirut, Lebanon

* myriam.abboud@zu.ac.ae

## Abstract

The prevalence of cardiometabolic abnormalities is high globally. This is concerning since these abnormalities increase the risk of morbidity and mortality. Using noninvasive, low-cost, and ethnic-specific anthropometric indices is crucial for widespread screening and early detection of cardiometabolic abnormalities. In this cross-sectional study, we enrolled 221 Lebanese participants (62.9% females; mean age: 43.36 ± 16.05 years; mean body mass index (BMI): 28.43 ± 6.10 Kg/m$^2$). The main outcome measure was cardiometabolic abnormality (CMA), defined as the presence of at least two or more non-anthropometric components of the Metabolic Syndrome. Several anthropometric indices: Total body fat percent, Conicity index, Abdominal volume index (AVI), Weight-adjusted-waist index, Waist circumference (WC), Neck circumference, Hip circumference, Waist-to-hip ratio, Waist-to-height ratio (WtHR), Neck-to-height ratio, and BMI were assessed in their prediction of CMA, using logistic regression modelling and c-statistic [95% confidence intervals (CIs)], and calibration plots, as well sensitivity, specificity, and negative and positive predictive values measures. The Benjamini-Hochberg correction procedure was used to correct for multiple testing. The prevalence of CMA was 52.0% (47.5% in females and 59.8% in males). Significant associations were found between all the anthropometric indices and CMA, except for neck-to-height ratio. AVI and WC were most predictive for CMA in the total sample. WtHR and WC were most predictive in females with suggested cut-off values of 0.58 and 91.25 cm, whereas AVI and WC were most predictive in males with suggested cut-off values of 19.61 and 101.50 cm. The neck-to-height measurement had the lowest predictive ability for CMA. Adding anthropometric indices to sociodemographic variables did not significantly improve model discrimination. AVI, WHtR, and WC best predicted CMA in a sample of Lebanese adults. These less invasive, low-cost, easy-to-measure indices can be used to

**Data Availability Statement:** All data underlying the findings described in their manuscript are uploaded as supplementary information.

**Funding:** This project was funded by Zayed University, United Arab Emirates under the Research Incentive Fund (RIF) (Grant number R21062 to MA, SH, NM, and RR). The funders had no role in study design, data collection and analysis, decision to publish, or preparation of the manuscript.

**Competing interests:** The authors have declared that no competing interests exist.

screen widely for CMA to better manage and prevent disease and subsequent morbidity and mortality.

## Introduction

Cardiometabolic abnormalities, i.e., hyperglycemia, elevated blood pressure, high triglyceride levels, low levels of high-density lipoprotein (HDL-C), and obesity [1] have long been associated with a higher risk of morbidity and mortality [2, 3].

Globally, the incidence of clustered cardiometabolic abnormalities, such as the metabolic syndrome (MetS), is rising due to alterations in the lifestyle, dietary choices, and stress exposure [4–7], with global estimates being 25% and ranging from 10–84% [8] depending on the diagnostic criteria used, age, gender, and ethnicity [9–11]. Early diagnosis of clustered cardiometabolic abnormalities is crucial for the adoption of lifestyle changes and lowering disease-related risk factors and optimizing treatment [12, 13], necessitating invasive and costly laboratory measurements of the plasma lipid and glycemic state [14].

Looking at the Middle Eastern region, the prevalence of the MetS has increased significantly in recent years [15]. This increase has been attributed to the westernization of the diet and increased intakes of fat, sugar, and salt [16]. Moreover, the prevalence has fluctuated based on the country and time of study with a pooled estimate of 25% between the years 2010–2014 [15, 17]. The Eastern Mediterranean region has the highest MetS prevalence across different MetS definitions that increased with the increase in the country's level of income [18]. Furthermore, in contrast to trends observed in northern Europe and the USA, mortality from cardiovascular diseases has risen, and which account for 25–45% of total deaths in these countries [19], with a higher prevalence amongst women than men [20]. Lebanon, a Middle-Eastern country, has one of the highest estimated prevalence of MetS among adults in the region (34.6%) [21]. The growing trend towards survival in later years, in addition to the high urbanization rates, and changes in lifestyles rendered the health of the Lebanese a major challenge [22].

Long standing evidence pinpoints excessive visceral fat as the main culprit behind the development of cardiometabolic abnormalities [23]. However, methods for measurement, such as computed tomography and magnetic resonance imaging are expensive and unavailable in routine clinical practice [24]. The use of straightforward anthropometric indices of body composition, such as body mass index (BMI), waist circumference (WC), and waist-to-height ratio (WHtR), has been traditionally used as a practical and non-invasive proxy for abdominal adiposity. However, studies found that the association of these indicators with clustered cardiometabolic abnormalities varies amongst populations [25]. More sensitive anthropometric indices that account for body shape may be better predictors of central obesity [23] as individuals with the same WC and BMI might share different risk factors based on differences in statures [26, 27]. Several anthropometric indices have been proposed as markers of body fat distribution and predictors of clustered cardiometabolic abnormalities, but without a thorough consensus concerning the optimal marker and its cut-off values [28–30].

Studies identifying the association between anthropometric indices and cardiometabolic risks in the Middle East region are scarce. Nasreddine et al. [31] identified the optimal cutoff values of three anthropometric indices and their relation to the MetS among Lebanese adults. The high incidence of cardiometabolic diseases in the region reinforces the need for further studies investigating the relation between several anthropometric indices and cardiometabolic risks. Furthermore, looking at ethnic and population-specific differences in the manifestation of cardiometabolic diseases in the region is warranted.

The purpose of this study is to examine the relationship between several anthropometric indices, both traditional and novel, and a cluster of cardiometabolic abnormalities in a sample of Lebanese adults, and to identify the optimal index and its cut-off values that best predict cardiometabolic abnormalities.

## Materials and methods

Lebanese adults were invited to participate in this cross-sectional study via community and university announcements. Interested participants were asked to be fasting for at least eight hours on the day of data collection. Inclusion criteria consisted of being aged between 18 and 65 years, of a Lebanese nationality, not diagnosed with a current infection, neither pregnant or lactating, and not being on medications nor suffering from diseases that interfere with vitamin D metabolism such as chronic kidney and liver diseases. Participants were asked about previous or current medical disease diagnosis. Data were collected between March and June 2022.

### Ethical considerations

The study was consistent with the declaration of Helsinki and approved by the Institutional Review Board of the Lebanese International University (Ethical approval no: LIUIRB-220201-SH-111).

All interested participants were approached by trained, qualified research assistants who explained them in detail the objectives of the study and the protocol. All participant were informed in detail about the study protocol and signed a consent form.

### Data collection

**Sociodemographic characteristics and medical history.**  A questionnaire was developed to collect data about demographics and medical history. It included questions about age, sex, education, employment, smoking history, and family history of diseases. Participants were asked to show their Lebanese identification card.

**Blood withdrawal.**  A 5 ml of blood was drawn by a licensed phlebotomist from participants in a fasting state after they had been resting for at least 5 minutes in a seated position. Blood was collected into a sterile serum separator tube with a clot activator and then transported to the laboratory for investigation via thermally insulated boxes after which the samples were centrifuged at 4000 revolutions per minute for 10 min and analyzed for blood lipids [total cholesterol (mg/dl), High-Density Lipoprotein-Cholesterol (HDL-C, mg/dl), triglycerides (mg/dl), fasting blood glucose (FBG)(mg/dl)], using an automated chemiluminescence micro-particleimmunoassay (CMIA) kit (ARCHITECT; Abbott Laboratories, Abbott Park, IL, USA). Low-density Lipoprotein-Cholesterol (LDL) was calculated using the Friedwald formula when triglyceride levels were below 400 mg/dl, otherwise they were measured.

**Blood Pressure (BP).**  After at least 5 minutes of rest and before blood withdrawal, a qualified technician used a standardized mercury sphygmomanometer to take two consecutive reading to measure systolic and diastolic blood pressure (SBP, DBP). Both readings were taken on the same arm where the average of both readings was used for analysis.

**Anthropometry.**  Anthropometric data was collected using standardized techniques and calibrated equipment by trained examiners. Participants were asked to take off their shoes and wear only light clothing during assessment. A portable stadiometer was used to measure height (ADE, Germany) and a beam scale was used to weigh the participants (kg). Height and weight were taken to the nearest 0.1 cm 100 g respectively while in participants were standing in the Frankfort plane. Body mass index (BMI) was calculated by dividing weight by squared height in meters ($m^2$). WC (cm) was measured to the nearest 0.1 cm at the mid-point, half-way

between the right iliac crest and the lower costal region using standardized measuring tape. Waist-to-height ratio (WHtR) was computed by dividing WC by height [32]. Neck circumference (NC-cm) was measured just below the laryngeal prominence perpendicular to the long axis of the neck while the head was positioned in Frankfurt horizontal plane [33]. Neck-to-heigh ratio was calculated by dividing neck circumference by height. Hip circumference (HC-cm) was measured using measuring tape at the level of the anterior superior iliac spine, at the widest circumference below the waist. Waist-to-hip ratio was then calculated by dividing the WC to hip circumference. Total body fat percent was measured using a multifrequency bioelectrical impedance analysis instrument (InBody 230, InBody, Seoul, Korea). For the bioelectrical impedance analysis, participants were fasting, with empty bladder, standing with both arms 45˚ apart from the body trunk, and having both feet bared on the spots of the equipment. Total body fat percent was shown on the equipment screen. Conicity index (C index), abdominal volume index (AVI), and weight-adjusted-waist index (WWI) were calculated as follows [14]:

$$C - Index = \frac{WC\ (m)}{0.109\sqrt{\frac{body\ weight\ (kg)}{height\ (m)}}}$$

$$AVI = \frac{2 \times (WC\ (cm))^2 + 0.7 \times (WC\ (cm) - HC(cm))^2}{1000}$$

$$WWI = \frac{WC\ (cm)}{\sqrt{weight\ (kg)}}$$

## Definition of clinical metabolic abnormality

A CMA was defined as the presence of two or more of the following non-anthropometric cardiometabolic abnormalities: elevated triglycerides ($\geq$150 mg/dL) or undergoing treatment for it; reduced HDL-C ($<$40 mg/dL in males and $<$50 mg/dL in females) or specific treatment for the abnormality; raised BP (SBP $\geq$130 or DBP$\geq$ 85 mmHg) or undergoing treatment for hypertension; and FBG $\geq$100 mg/dL or previously diagnosed with diabetes.

## Statistical analysis

The data were analyzed using the SPSS software, version 25. A descriptive analysis was done using the counts and percentages for categorical variables (sociodemographic characteristics and the presence/absence of CMA) and mean and standard deviation for continuous measures. The sample was normally distributed, as checked by visual inspection of the histogram of the independent variables, i.e., the anthropometric indices.

The Chi-square and Fisher exact tests were used to assess associations between categorical variables and CMA, and the student t-test was used to assess associations between continuous variables and CMA. The Analysis of Covariance (ANCOVA) was used to display the difference in mean of the anthropometric indices between presence and absence of CMA adjusted over covariates (age, sex, education level, socioeconomic status, smoking, family history of diabetes, family history of dyslipidemia, and family history of hypertension). To correct for multiple testing the Benjamini-Hochberg method was used for the analyses in the total sample and then in each sex separately (three groups of analyses with 11 tests performed in each group) [34]. All the other associations that have a p-values larger than the critical values are considered not

significant. Logistic regression models and c statistic calculations were performed taking the presence/absence of CMA as the dependent variable and anthropometric indices as independent variables, adjusted over the sociodemographic and lifestyle characteristics, and family history of diseases. The c statistic and 95% confidence intervals (CIs) was calculated by using the ROC curve analysis and was used to assess the discriminatory power of each anthropometric indices to assess the risk for CMA. The calibration plot was used to assess calibration. It represents graphically the agreement between predicted and actual probabilities. It is based on the contingency table for Hosmer-Lemeshow statistic, which aggregates cases into 10 groups of approximately equal size established according to their predicted values arranged from lowest to highest values. A linear regression model was applied to estimate the calibration slope and draw the line that best fitted the points in the calibration plot. The cut-off values were determined by using the values of the ROC-curve of each anthropometric measure apart, where the highest sensitivity values and specificity were considered to choose the best values. Also, the sensitivity and specificity as well as the negative and positive predictive values were calculated to assess the probability of the measures to indicate the CMA. Since the anthropometric indices are correlated with sex, the analysis was stratified into two groups (males and females). P-values < 0.05 were considered statistically significant, except when adjusting over multiple testing.

## Results

### Sample description

A total of 221 participants agreed to enroll in this study. Most of the participants were females (62.9%) and more than half of them were married (55.7%) with low socioeconomic income (50.2%). Almost half of them have a university education level (46.2%) and were employed (46.6%). Only 21.3% were cigarette smokers and 31.7% smoke waterpipe. Regarding diseases distribution, 18.1% had diabetes, 28.5% had dyslipidemia, and 20.8% had hypertension. More than half of the participants have a family history of diabetes (54.3%) and hypertension (57.0%), and 38.5% had a family history of dyslipidemia. The mean age of the participants was 43.36 ± 16.05 years. Table 1 shows the sociodemographic and other characteristics of the participants.

More than half of the sample (52.0%) presented CMA; this was more common in males than females (59.8% vs. 47.5%, respectively). The association between the anthropometric indices and CMA is presented in Table 2. The results showed a significant association between all the measures and CMA, in the total sample except for the neck-to-height ratio. When stratifying by sex, whereby significantly higher mean anthropometric indices were found in those having CMA as compared with those who do not have CMA except for the total body fat, neck and hip circumference and neck-to-heigh ratio among females. However, among males, significantly higher mean anthropometric indices were found in those having CMA as compared with those who do not have CMA except for the C-index, the WWI, waist-to-hip ratio and the neck-to-height ratio.

As per Table 3, in the total sample and in females, when considering the sociodemographic characteristics as independent variables, the results showed that higher age and having a university education level were significantly associated with the presence of CMA. Among males, higher age was significantly associated with the presence of CMA. As for the anthropometric indices, among the total sample and in males, all the measures were significantly associated with the presence of CMA, except the neck-to-height ratio. Among females, a higher C-index, AVI, WWI, WC, waist-to-hip ratio, WtHR, and BMI were significantly associated with the presence of CMA.

**Table 1. Sociodemographic and other characteristics of the participants (N = 221).**

| Characteristics | Missing values N (%) | Total sample | Females | Males |
|---|---|---|---|---|
| **General characteristics** | | | | |
| **Mean age in years** | 1 (0.5%) | 43.36 (SD, 16.05) | 42.69 (SD, 16.00) | 44.53 (SD, 16.17) |
| **Female** | 0 | 139 (62.9%) | - | - |
| **Marital status** (Married) | 0 | 123 (55.7%) | 65 (46.8%) | 58 (70.7%) |
| **Education level** | | | | |
| University degree | 0 | 102 (46.2%) | 59 (42.4%) | 43 (52.4%) |
| High school | 0 | 41 (18.6%) | 29 (20.9%) | 12 (14.6%) |
| Middle education | 0 | 37 (16.7%) | 25 (18.0%) | 12 (14.6%) |
| Primary education | 0 | 30 (13.6%) | 15 (10.8%) | 15 (18.3%) |
| Illiterate | 0 | 11 (5.0%) | 11 (7.9%) | - |
| **Socioeconomic status** | 1 (0.5%) | | | |
| Low | | 111 (50.2%) | 72 (52.2%) | 39 (47.6%) |
| Medium | | 102 (46.2%) | 63 (45.7%) | 39 (47.6%) |
| High | | 7 (3.2%) | 3 (2.2%) | 4 (4.9%) |
| **Employment status** (employed) | 2 (0.9%) | 103 (46.6%) | 46 (33.6%) | 57 (69.5%) |
| **Cigarette smoker** | 0 | 47 (21.3%) | 27 (19.4%) | 20 (24.4%) |
| **Waterpipe smoker** | 0 | 70 (31.7%) | 49 (35.3%) | 21 (25.6%) |
| **Cardiometabolic abnormality (CMA)** | 0 | 115 (52.0%) | 66 (47.5%) | 49 (59.8%) |
| **Comorbidities** | | | | |
| Personal history of diabetes | 0 | 40 (18.1%) | 26 (18.7%) | 14 (17.1%) |
| Family history of diabetes | 1 (0.5%) | 120 (54.3%) | 76 (55.1%) | 44 (53.7%) |
| Personal history of dyslipidemia | 0 | 63 (28.5%) | 43 (30.9%) | 20 (24.4%) |
| Family history of dyslipidemia | 1 (0.5%) | 85 (38.5%) | 58 (42.0%) | 27 (32.9%) |
| Personal history of hypertension | 0 | 46 (20.8%) | 27 (19.4%) | 19 (23.2%) |
| Family history of hypertension | 1 (0.5%) | 126 (57.0%) | 83 (59.7%) | 43 (53.1%) |
| **Anthropometric indices (mean)** | | | | |
| Total body fat percent | 0 | 35.54 (SD, 10.28) | 40.41 (SD, 7.96) | 27.30 (SD, 8.37) |
| Conicity index | 0 | 1.29 (SD, 0.13) | 1.27 (SD, 0.15) | 1.33 (SD, 0.09) |
| Abdominal volume index | 8 (3.6%) | 19.42 (SD, 6.15) | 18.68 (SD, 6.41) | 20.66 (SD, 5.50) |
| Weight-adjusted-waist index | 0 | 0.11 (SD, 0.01) | 0.11 (SD, 0.01) | 0.11 (SD, 0.08) |
| Waist circumference (cm) | 0 | 96.31 (SD, 16.58) | 93.92 (SD, 17.60) | 100.37 (SD, 13.88) |
| Neck circumference (cm) | 0 | 40.40 (SD, 4.35) | 39.26 (SD, 4.08) | 42.35 (SD, 4.12) |
| Hip circumference (cm) | 8 (3.6%) | 107.93 (SD, 12.73) | 108.75 (SD, 13.69) | 106.56 (SD, 10.88) |
| Waist-to-hip ratio | 8 (3.6%) | 0.89 (SD, 0.12) | 0.87 (SD, 0.14) | 0.94 (SD, 0.08) |
| Waist-to-height ratio | 0 | 0.58 (SD, 0.10) | 0.59 (SD, 0.11) | 0.58 (SD, 0.08) |
| Neck-to-height ratio | 0 | 0.24 (SD, 0.02) | 0.24 (SD, 0.02) | 0.24 (SD, 0.02) |
| Body Mass Index (BMI) | 0 | 28.43 (SD, 6.10) | 28.86 (SD, 6.57) | 27.69 (SD, 5.15) |

**Table 2. Association between anthropometric indices and clinical metabolic abnormality (CMA) in the total sample and both sexes (N = 221).**

| | Total sample | | | Females | | | Males | | |
|---|---|---|---|---|---|---|---|---|---|
| | Presence of CMA (n = 115) | Absence of CMA (n = 106) | | Presence of CMA (n = 66) | Absence of CMA (n = 73) | | Presence of CMA (n = 49) | Absence of CMA (n = 33) | |
| | Mean (SE) | Mean (SE) | MD (95% CI) | Mean (SE) | Mean (SE) | MD (95% CI) | Mean (SE) | Mean (SE) | MD (95% CI) |
| **Total body fat percent** | 37.53 (0.75) | 33.33 (0.77) | 4.19 (1.87; 6.52) | 41.89 (1.05) | 38.86 (0.95) | 3.03 (-0.12; 6.18) | 29.22 (1.14) | 24.61 (1.40) | 4.62 (0.75; 8.48) |
| p-value | **<0.001** | | | 0.042 | | | **0.017** | | |
| **Conicity index** | 1.33 (0.01) | 1.25 (0.01) | 0.07 (0.04; 0.11) | 1.32 (0.02) | 1.23 (0.01) | 0.09 (0.03; 0.15) | 1.35 (0.01) | 1.31 (0.01) | 0.04 (0.002; 0.07) |
| p-value | **<0.001** | | | **0.007** | | | 0.074 | | |
| **Abdominal volume index** | 21.85 (0.55) | 16.79 (0.56) | 5.05 (3.36; 6.74) | 21.02 (0.80) | 16.44 (0.74) | 4.58 (2.13; 7.02) | 22.66 (0.75) | 17.96 (0.90) | 4.69 (2.17; 7.21) |
| p-value | **<0.001** | | | **0.001** | | | **0.001** | | |
| **Weight-adjusted-waist index** | 0.11 (0.001) | 0.10 (0.001) | 0.006 (0.03; 0.01) | 0.11 (0.002) | 0.10 (0.002) | 0.008 (0.003; 0.01) | 0.11 (0.001) | 0.10 (0.001) | 0.002 (-0.001; 0.006) |
| p-value | **0.001** | | | **0.009** | | | 0.190 | | |
| **Waist circumference (cm)** | 102.53 (1.44) | 89.53 (1.47) | 12.99 (8.54; 17.45) | 100.38 (2.16) | 87.87 (1.97) | 12.51 (6.00; 19.02) | 104.90 (1.84) | 93.93 (2.26) | 10.96 (4.70; 17.21) |
| p-value | **<0.001** | | | **0.001** | | | **0.001** | | |
| **Neck circumference (cm)** | 41.14 (0.41) | 39.75 (0.42) | 1.39 (0.13; 2.65) | 39.49 (0.57) | 39.09 (0.52) | 0.40 (-1.31; 2.11) | 43.56 (0.62) | 40.92 (0.76) | 2.64 (0.54; 4.73) |
| p-value | **0.034** | | | 0.765 | | | **0.015** | | |
| **Hip circumference (cm)** | 110.84 (1.28) | 104.55 (1.31) | 6.29 (2.33; 10.25) | 110.16 (1.95) | 106.85 (1.81) | 3.30 (-2.64; 9.26) | 109.97 (1.57) | 102.16 (1.88) | 7.81 (2.53; 13.09) |
| p-value | **0.001** | | | 0.300 | | | **0.003** | | |
| **Waist-to-hip ratio** | 0.93 (0.01) | 0.85 (0.01) | 0.07 (0.03; 0.11) | 0.92 (0.02) | 0.82 (0.02) | 0.10 (0.04; 0.16) | 0.95 (0.01) | 0.91 (0.01) | 0.04 (-0.003; 0.08) |
| p-value | **<0.001** | | | **0.002** | | | 0.145 | | |
| **Waist-to-height ratio** | 0.62 (0.09) | 0.55 (0.09) | 0.07 (0.04; 0.10) | 0.63 (0.01) | 0.55 (0.01) | 0.07 (0.03; 0.12) | 0.60 (0.01) | 0.55 (0.01) | 0.05 (0.01; 0.09) |
| p-value | **<0.001** | | | **0.001** | | | **0.004** | | |
| **Neck-to-height ratio** | 0.25 (0.003) | 0.24 (0.003) | 0.007 (-0.001; 0.01) | 0.24 (0.004) | 0.24 (0.003) | 0.003 (-0.008; 0.01) | 0.25 (0.004) | 0.24 (0.005) | 0.01 (-0.001; 0.02) |
| p-value | 0.085 | | | 0.775 | | | 0.050 | | |
| **Body Mass Index** | 30.35 (0.57) | 26.25 (0.58) | 4.09 (2.33; 5.85) | 30.54 (0.85) | 27.14 (0.77) | 3.40 (0.84; 5.95) | 29.42 (0.73) | 25.24 (0.89) | 4.17 (1.69; 6.65) |
| p-value | **<0.001** | | | **0.018** | | | **0.001** | | |

CMA: Clinical Metabolic Abnormality; SE: Standard Error; MD: Mean Difference; CI: Confidence Interval

The results were adjusted over age, sex, education level, socioeconomic status, smoking, family history of diabetes, family history of dyslipidemia, and family history of hypertension.

To correct for multiple testing the Benjamini-Hochberg correction procedure was used. The following formula was used to calculate the p-values (i/m)Q where i is the individual p-value's rank, m = total number of tests and Q = the false discovery rate (0.05). The values marked in Bold were considered as significant.

When considering the c-statistic, the highest difference for men was 0.85–0.77 = 0.08, while for women it was 0.89–0.86 = 0.03. In the total sample, the highest c-statistic was found for AVI (0.88 (0.83; 0.92)) and WC (0.88 (0.83; 0.92)), however the lowest values were found for neck-to-height ratio (0.84 (0.78; 0.89)) and NC (0.84 (0.79; 0.89)). Regarding the sociodemographic variables, the c-statistic was 0.83; when adding the anthropometric measure, the c-statistic improved to 0.84 and 0.88.

**Table 3. Logistic regression analysis taking the presence of CMA in the total sample and both sexes as the dependent variable.**

| | | Dependent variable: presence vs absence of CMA | | | | | | | | |
|---|---|---|---|---|---|---|---|---|---|---|
| | | Total sample (N = 212, missing 9 (4.1%)) | | | Females (N = 133, missing 6 (4.3%)) | | | Males (N = 79, missing 3 (3.7%)) | | |
| | | ORa (95% CI) | c statistic (95% CI) | p-value | ORa (95% CI) | c statistic (95% CI) | p-value | ORa (95% CI) | c statistic (95% CI) | p-value |
| | **Independent variables** | | | | | | | | | |
| | **Sociodemographic variables** | | | | | | | | | |
| Model 1 | Age | 1.07 (1.05; 1.10) | 0.83 (0.77; 0.88) | **<0.001** | 1.08 (1.04; 1.11) | 0.86 (0.80; 0.92) | **<0.001** | 1.09 (1.04; 1.15) | 0.77 (0.66; 0.87) | **<0.001** |
| | Sex (female vs male*) | 2.05 (1.00; 4.20) | | 0.049 | - | | | - | | - |
| | Education level (university level vs illiterate*) | 0.32 (0.16; 0.64) | | **0.001** | 0.18 (0.07; 0.45) | | **<0.001** | 0.49 (0.14; 1.70) | | 0.267 |
| | Marital status (married vs single*) | 0.99 (0.48; 2.06) | | 0.992 | 1.93 (0.77; 4.79) | | 0.156 | 0.20 (0.03; 1.11) | | 0.066 |
| | Smoking status (yes vs no*) | 0.94 (0.48; 1.83) | | 0.865 | 1.09 (0.44; 2.65) | | 0.849 | 0.89 (0.29; 2.73) | | 0.848 |
| | Family history of diabetes (yes vs no*) | 1.02 (0.51; 2.03) | | 0.952 | 0.91 (0.35; 2.34) | | 0.847 | 1.45 (0.48; 4.43) | | 0.506 |
| | Family history of dyslipidemia (yes vs no*) | 1.47 (0.71; 3.03) | | 0.291 | 1.67 (0.65; 4.26) | | 0.283 | 0.97 (0.27; 3.49) | | 0.964 |
| | Family history of hypertension (yes vs no*) | 1.52 (0.75; 3.10) | | 0.241 | 1.66 (0.63; 4.37) | | 0.299 | 1.55 (0.49; 4.87) | | 0.452 |
| | **Anthropometric indices**** | | | | | | | | | |
| Model 2 | Total body fat percent | 2.52 (1.53; 4.14) | 0.85 (0.80; 0.90) | **<0.001** | 2.07 (1.07; 4.00) | 0.87 (0.81; 0.93) | **0.029** | 3.78 (1.59; 8.95) | 0.82 (0.73; 0.91) | **0.002** |
| Model 3 | Conicity index | 2.76 (1.63; 4.68) | 0.85 (0.80; 0.90) | **<0.001** | 2.43 (1.30; 4.55) | 0.88 (0.82; 0.94) | **0.005** | 3.75 (1.29; 10.86) | 0.80 (0.70; 0.90) | **0.015** |
| Model 4 | Abdominal volume index | 3.16 (2.01; 4.94) | 0.88 (0.83; 0.92) | **<0.001** | 2.69 (1.54; 4.72) | 0.88 (0.83; 0.94) | **0.001** | 4.87 (2.03; 11.69) | 0.85 (0.77; 0.94) | **<0.001** |
| Model 5 | Weight-adjusted-waist index | 2.58 (1.53; 4.36) | 0.85 (0.80; 0.90) | **<0.001** | 2.45 (1.31; 4.58) | 0.88 (0.82; 0.94) | **0.005** | 3.02 (1.00; 9.09) | 0.79 (0.68; 0.89) | **0.048** |
| Model 6 | Waist circumference | 3.23 (2.04; 5.12) | 0.88 (0.83; 0.92) | **<0.001** | 2.75 (1.56; 4.84) | 0.89 (0.84; 0.94) | **<0.001** | 5.04 (2.04; 12.44) | 0.85 (0.76; 0.93) | **<0.001** |
| Model 7 | Neck circumference | 1.63 (1.12; 2.36) | 0.84 (0.79; 0.89) | **0.010** | 1.41 (0.85; 2.32) | 0.87 (0.80; 0.93) | 0.173 | 2.05 (1.06; 3.96) | 0.79 (0.69; 0.89) | **0.031** |
| Model 8 | Hip circumference | 1.91 (1.28; 2.85) | 0.85 (0.80; 0.90) | **0.001** | 1.46 (0.95; 2.23) | 0.86 (0.79; 0.92) | 0.078 | 8.69 (2.63; 28.76) | 0.82 (0.73; 0.91) | **<0.001** |
| Model 9 | Waist-to-hip ratio | 2.76 (1.56; 4.90) | 0.85 (0.80; 0.90) | **<0.001** | 2.74 (1.32; 5.68) | 0.88 (0.82; 0.93) | **0.007** | 3.05 (1.02; 9.14) | 0.79 (0.70; 0.89) | **0.046** |
| Model 10 | Waist-to-height ratio | 3.10 (1.95; 4.93) | 0.87 (0.82; 0.92) | **<0.001** | 2.71 (1.55; 4.71) | 0.89 (0.83; 0.94) | **<0.001** | 4.96 (1.87; 13.13) | 0.83 (0.74; 0.92) | **0.001** |
| Model 11 | Neck-to-height ratio | 1.53 (1.06; 2.20) | 0.84 (0.78; 0.89) | **0.022** | 1.43 (0.88; 2.31) | 0.87 (0.81; 0.93) | 0.141 | 1.73 (0.90; 3.33) | 0.78 (0.68; 0.88) | 0.095 |
| Model 12 | Body Mass Index | 2.49 (1.64; 3.79) | 0.86 (0.81; 0.91) | **<0.001** | 2.00 (1.22; 3.26) | 0.87 (0.81; 0.93) | **0.005** | 5.18 (2.00; 13.42) | 0.84 (0.75; 0.92) | **0.001** |

ORa: adjusted Odds Ratio, CI: Confidence interval

*reference category

**The models were adjusted over the sociodemographic characteristics: age, sex, education level, marital status, smoking status, family history of diabetes, family history of dyslipidemia, and family history of hypertension

Among females, the highest c-statistic was found for WHtR (0.89 (0.83; 0.94)) and WC (0.89 (0.84; 0.94)), however the lowest values were found for HC (0.86 (0.79; 0.92)).

Among males, the highest c-statistic was found for AVI (0.85 (0.77; 0.94)) and WC (0.85 (0.76; 0.93)), however the lowest values were found for neck-to-height ratio (0.78 (0.68; 0.88)). When adding the anthropometric indices to the model with the sociodemographic variables, the discrimination did not significantly improve in the total sample or analyses stratified by sex. The discrimination power of the sociodemographic variables and the anthropometric indices to predict CMA in the total sample and between sexes are shown in Fig 1.

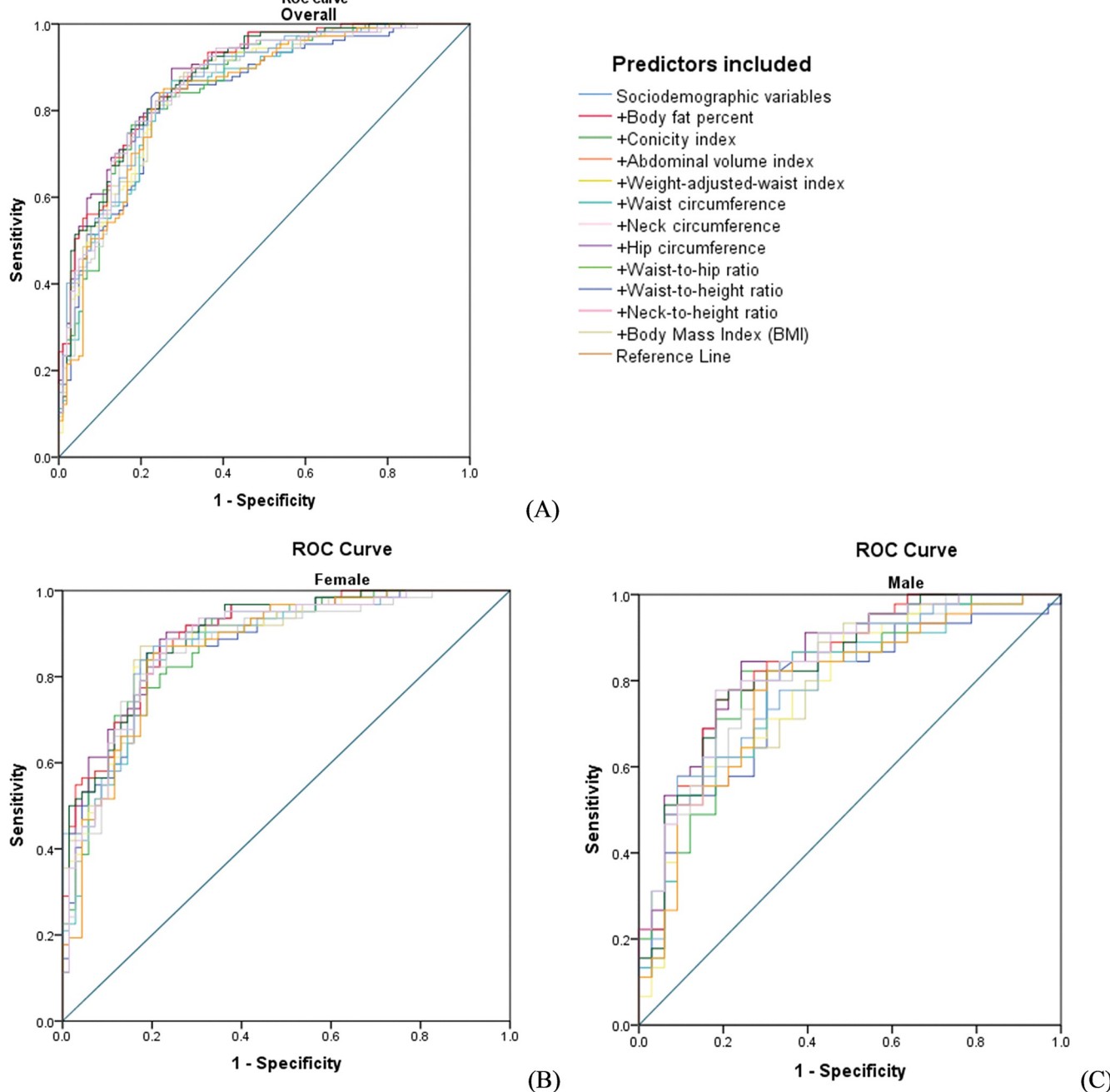

**Fig 1.** ROC-curves illustrating the discriminatory power of the different anthropometric indices to predict CMA in the total sample (A), among females (B), and among males (C).

The c-statistic for each of the anthropometric indices for predicting the CMA in total sample and both sexes is presented in the supplementary material (S1 Table).

As for the calibration analysis between the expected and observed cases to estimate the presence of CMA, the ratio between the expected and observed cases showed an adequate calibration and the respective calibration plot showed no significant over or underestimation of predictor effects in total sample (Fig 2) and in both sexes (S1, S2 Figs and S2 Table).

Table 4 shows the measures of diagnostic accuracy of each of the anthropometric indices for the presence of CMA in the total sample and in both sexes. In the total sample, the results showed optimal cut-off values of 18.88 for AVI, 0.58 for WHtR, and 94.5 cm for WC. Among females, optimal cut-off values were 16.75 for AVI, 0.58 for WHtR, and 91.25 cm for WC, and among males, optimal cut-off values were 19.61 for AVI, 0.56 for WHtR, and 101.50 cm for WC.

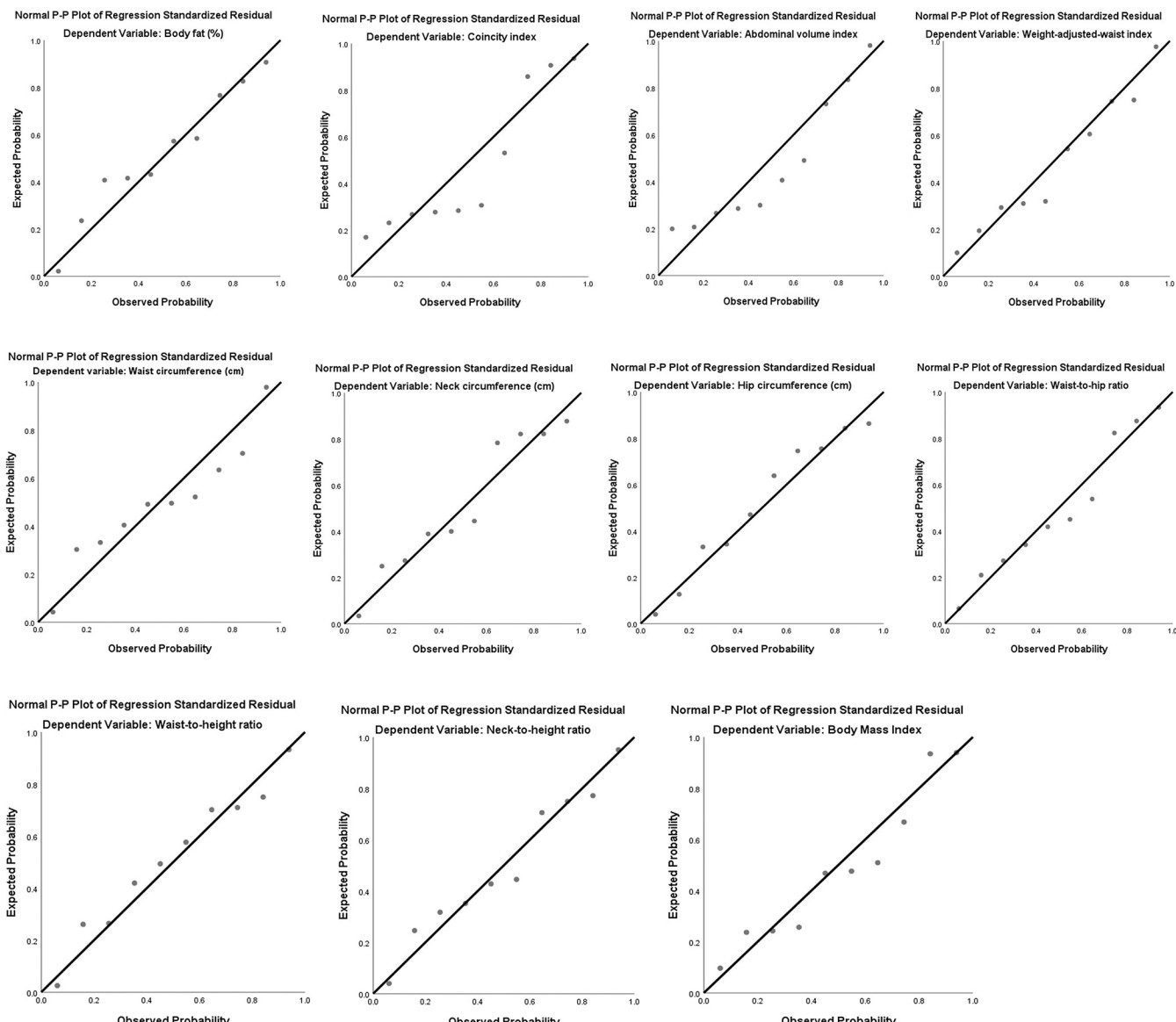

**Fig 2. Calibration plots of the anthropometric measures based on the contingency table for Hosmer-Lemeshow statistic in total sample.**

**Table 4. Measures of diagnostic accuracy of each anthropometric indices for the presence of CMA in the total sample and both sexes (N = 221).**

**Total sample**

| | Value | Sensitivity | Specificity | PPV | NPV |
|---|---|---|---|---|---|
| Total body fat percent | 34.35 | 70.30 | 55.90 | 62.76 | 62.86 |
| Conicity index | 1.28 | 80.20 | 61.80 | 68.97 | 73.79 |
| Abdominal volume index | 18.88 | 74.80 | 71.60 | 73.44 | 71.60 |
| Weight-adjusted-waist index | 0.10 | 80.20 | 62.70 | 69.52 | 74.10 |
| Waist circumference (cm) | 94.50 | 77.50 | 70.60 | 74.06 | 74.34 |
| Neck circumference (cm) | 39.60 | 73.00 | 52.90 | 62.67 | 64.39 |
| Hip circumference (cm) | 106.75 | 70.30 | 60.80 | 66.02 | 65.39 |
| Waist-to-hip ratio | 0.87 | 79.30 | 60.80 | 68.67 | 73.05 |
| Waist-to-height ratio | 0.58 | 77.50 | 70.60 | 74.06 | 74.34 |
| Neck-to-height ratio | 0.23 | 70.30 | 53.90 | 62.29 | 62.62 |
| Body Mass Index | 27.86 | 72.10 | 70.60 | 72.65 | 70.02 |
| **Females** | | | | | |
| Total body fat percent | 40.20 | 73.40 | 62.30 | 63.01 | 72.14 |
| Conicity index | 1.27 | 73.40 | 63.80 | 63.63 | 72.46 |
| Abdominal volume index | 16.75 | 81.30 | 72.50 | 71.95 | 81.04 |
| Weight-adjusted-waist index | 0.11 | 70.30 | 68.10 | 65.98 | 71.88 |
| Waist circumference (cm) | 91.25 | 81.30 | 73.90 | 73.81 | 81.37 |
| Neck circumference (cm) | 38.25 | 67.20 | 52.20 | 55.99 | 63.75 |
| Hip circumference (cm) | 106.50 | 73.40 | 60.90 | 62.94 | 71.68 |
| Waist-to-hip ratio | 0.86 | 70.30 | 66.70 | 65.64 | 71.28 |
| Waist-to-height ratio | 0.58 | 81.30 | 72.50 | 72.79 | 81.08 |
| Neck-to-height ratio | 0.24 | 70.30 | 56.60 | 59.44 | 67.81 |
| Body Mass Index | 28.14 | 81.30 | 72.50 | 72.79 | 81.08 |
| **Males** | | | | | |
| Total body fat percent | 26.50 | 70.20 | 60.60 | 71.58 | 58.16 |
| Conicity index | 1.32 | 70.20 | 63.60 | 73.14 | 59.34 |
| Abdominal volume index | 19.61 | 76.60 | 63.60 | 74.72 | 64.59 |
| Weight-adjusted-waist index | 0.11 | 70.20 | 66.70 | 74.76 | 60.46 |
| Waist circumference (cm) | 101.50 | 70.20 | 63.60 | 74.15 | 58.93 |
| Neck circumference (cm) | 41.75 | 68.10 | 57.60 | 70.49 | 54.83 |
| Hip circumference (cm) | 105.75 | 72.30 | 60.60 | 73.19 | 59.53 |
| Waist-to-hip ratio | 0.92 | 80.90 | 57.60 | 73.95 | 66.97 |
| Waist-to-height ratio | 0.56 | 83.00 | 60.60 | 75.81 | 70.56 |
| Neck-to-height ratio | 0.23 | 70.20 | 51.50 | 68.29 | 53.74 |
| Body Mass Index | 27.11 | 70.20 | 69.70 | 77.51 | 61.12 |

## Discussion

In a sample of Lebanese adults, we found an alarmingly high prevalence of CMA reaching approximately 52%, with males having a notably higher prevalence (60%) than females (47.5%). There was a significant association of all anthropometric indices- both traditional and novel, with CMA except for the neck to height ratio. AVI and WC were the most predictive anthropometric indices in the study sample, whereas neck-to-height ratio and NC had the least predictive ability. Among females, WHtR and WC were most predictive of CMA, whereas HC was found to be the least predictive. Among males, the best anthropometric predictor for CMA was AVI followed by WC whereas the neck-to-height ratio had the least predictive

ability Finally, adding anthropometric indices to models with sociodemographic variables did not significantly improve the discriminatory ability of the model.

We found AVI to be a good predictor for CMA in the total sample, at an optimal cut off point of 19.61 and 16.75 among males and females, respectively. Recently, Al-Ahmadi et al. [14] also identified AVI as a marker for CMA in Saudi adults and identified similar cut-offs to those calculated in this study (19.1 for males and 16.3 for females). Similarly, WHtR was identified by Khader et al. [35] as one of the best predictors for metabolic abnormalities among adult Jordanians, with a similar cut-off value of 0.5 compared with the 0.56 as computed in our study. Recent data also support AVI and WHtR as optimal predictors for metabolic abnormalities even in non-overweight/obese adults [36]. Of clinical importance, a WHtR of 0.5 is emerging as a cut-off for metabolic abnormalities among different ethnicities and even in non-overweight/obese adults [36]. The value of a WHtR of 0.5 as a universal cut-off for screening for metabolic abnormalities is confirmed by the meta-analysis by Ashwell et al. [37], and large epidemiological studies in several countries [38, 39]. It has been hypothesized that WHtR is a more reliable predictor than WC alone given the genetic advantage of height on cardiometabolic risk whether it be due to genetics alone or the effect of intrauterine exposures during critical period of growth and development [37]. Hence, indices adjusting WC over height, i.e., AVI and WHtR, present valuable screeners for metabolic abnormalities, given their ease of measurement requiring low-cost equipment (measuring tape and stadiometer) which are widely available and that could easily be applied by all health care practitioners, in different settings including the community, primary health care, and clinical settings. Nevertheless, given the greater simplicity of its calculation, we suggest the use of WHtR in routine practice.

Furthermore, in our study sample and among stratified sexes, WC was a good predictor of CMA, with optimal thresholds of 91.25 cm in females and 101.5 cm in males. Similar cut-off values for adult Lebanese were also reported in a national epidemiological study (91.5 cm in females and 98.5 cm in males) [40]. These results suggest that optimal central obesity cut-off points in Lebanon may be higher than those recommended by the International Diabetes Federation (IDF), i.e., 80 cm in females and 94 cm in males [1] justifying the need for nation-wide studies to elucidate the optimal cut-off to be used in screening protocols. Data from neighboring countries also suggest higher cut-off values for predicting cardiometabolic abnormalities [41, 42]. All of which questions the current use of Europids criterion for WC in the diagnosis of the MetS in adult Arabs.

Clinicians have long been using BMI as a proxy measure for adiposity and screener for cardiometabolic risk [43]. Our results do not support the routine use of BMI for this aim, considering the more reliable indices that we have identified, especially since BMI does not account for muscle mass nor does it identify whether excess fat is accumulating in the abdominal region [44]. Existing meta-analyses including studies conducted among adults from various ethnicities have also questioned the clinical relevance of BMI, and support the use of WHtR in identifying adults at increased cardiometabolic risk [37].

Our finding of total body fat percent did not have predictive ability for CMA in females also challenges its clinical screening relevance. This is an interesting finding since clinicians often rely on measurement of total body fat to confer risk of cardiometabolic abnormalities. The distribution of body fat matters more than total body fat when related to cardiometabolic risk, as abdominal obesity plays a central role in the development of cardiometabolic abnormalities [45]. Suggested pathways through which visceral fat negatively impacts the metabolic profile include production of inflammatory mediators and increased rate of fatty acid release [46]. Hence, our results, in line with available literature, do not support the clinical use of total body fat percent in screening for cardiometabolic risk; rather, if body fat will be used, segmental analysis may be better to opt for. Nevertheless, since analyzers of visceral fat vary in quality

and units of measurement, further studies are needed to elucidate the optimal index and respective cut-off value to be used in clinical practice.

To our knowledge this is one of the few studies that have been conducted in the Middle Eastern region to identify novel ethnic specific anthropometric indices in their prediction of cardiometabolic abnormalities. Additionally, all the instruments used in this study have been validated. Furthermore, the research team recruited was highly qualified and adequately trained to collect data. Finally, total body fat percent was measured by a bioelectrical impedance analyzer which results significantly correlated with those by dual-energy X-ray absorptiometry (DXA) [47]. However, our study was not without limitations one of which was the recruitment of the participants who responded to community announcement and thus a risk of self-selection bias could exist. Additionally, this was not a national study, and was conducted on a relatively small sample, so the results, especially the reported prevalence and cut-off points, might not be generalizable to the general Lebanese population Hence future studies should be conducted to recruit a more representative and bigger sample to ensure adequate cut-offs. Future studies should also account for potential predictors not available in this study, including alcohol, diet, and physical activity.

In conclusion, our study suggests WtHR and WC as most predictive of CMA in females with suggested cut-off values of 0.58 and 91.25 cm, and AVI and WC as most predictive in males with suggested cut-off values of 19.61 and 101.50 cm. We recommend the use of these less invasive, low-cost, easy-to-measure indices to screen widely for CMA to better manage and prevent disease and subsequent morbidity and mortality. To enable clinical utilization by healthcare providers, it is necessary to conduct future cohort prospective studies that encompass a broader range of variables and ensure national representativeness. These studies will yield cutoff values that can be employed effectively.

## Supporting information

**S1 Fig. Calibration plot of the anthropometric measures based on the contingency table for Hosmer-Lemeshow statistic among females.**
(DOCX)

**S2 Fig. Calibration plot of the anthropometric measures based on the contingency table for Hosmer-Lemeshow statistic among males.**
(DOCX)

**S1 Table. The area under the curve of each anthropometric indices for the presence of CMA in the total sample and both genders (N = 221).**
(DOCX)

**S2 Table. Calibration in the presence of CMA in the overall sample and stratified by sex.**
(DOCX)

**S1 Data.**
(SAV)

## Author Contributions

**Conceptualization:** Myriam Abboud, Dimitrios Papandreou, Rana Rizk.

**Data curation:** Suzan Haidar.

**Formal analysis:** Suzan Haidar, Nadine Mahboub, Rana Rizk.

**Funding acquisition:** Myriam Abboud, Dimitrios Papandreou.

**Investigation:** Rana Rizk.

**Methodology:** Myriam Abboud, Suzan Haidar, Nadine Mahboub.

**Project administration:** Myriam Abboud.

**Supervision:** Myriam Abboud, Dimitrios Papandreou.

**Validation:** Dimitrios Papandreou.

**Visualization:** Nadine Mahboub, Dimitrios Papandreou.

**Writing – original draft:** Myriam Abboud, Nadine Mahboub, Rana Rizk.

**Writing – review & editing:** Myriam Abboud.

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
