## [Decision Letter · Decision Letter 0]

4 Jul 2023

PGPH-D-23-00803

Abdominal volume index, waist-to-height ratio, and waist circumference are optimal predictors of cardiometabolic abnormalities in a sample of Lebanese adults: A cross-sectional study

Dear Dr. Abboud,

Thank you for submitting your manuscript to PLOS Global Public Health. After careful consideration, we feel that it has merit but does not fully meet PLOS Global Public Health’s publication criteria as it currently stands. Therefore, we invite you to submit a revised version of the manuscript that addresses the points raised during the review process.

We look forward to receiving your revised manuscript.

Kind regards,

Derrick Anthony Bennett

Academic Editor

Journal Requirements:

1. Please send a completed 'Competing Interests' statement, including any COIs declared by your co-authors. If you have no competing interests to declare, please state "The authors have declared that no competing interests exist". Otherwise please declare all competing interests beginning with twhe statement "I have read the journal's policy and the authors of this manuscript have the following competing interests:"

3. Please provide separate figure files in .tif or .eps format only and remove any figures embedded in your manuscript file. Please also ensure all files are under our size limit of 10MB.

4. In the online submission form, you indicated that "All data are available and kept safely and confidentially with corresponding author". All PLOS journals now require all data underlying the findings described in their manuscript to be freely available to other researchers, either 1. In a public repository, 2. Within the manuscript itself, or 3. Uploaded as supplementary information.

Additional Editor Comments (if provided):

The authors need to adhere to the appropriate reporting guidelines when revising the manuscript.

Greater clarity in the reporting of the statistical analyses are required.

Reviewers' comments:

Reviewer's Responses to Questions

**Comments to the Author**

1. Does this manuscript meet PLOS Global Public Health’s publication criteria? Is the manuscript technically sound, and do the data support the conclusions? The manuscript must describe methodologically and ethically rigorous research with conclusions that are appropriately drawn based on the data presented.

Reviewer #1: Yes

Reviewer #2: Partly

2. Has the statistical analysis been performed appropriately and rigorously?

Reviewer #1: Yes

Reviewer #2: No

3. Have the authors made all data underlying the findings in their manuscript fully available (please refer to the Data Availability Statement at the start of the manuscript PDF file)?

Reviewer #1: Yes

Reviewer #2: No

4. Is the manuscript presented in an intelligible fashion and written in standard English?

Reviewer #1: Yes

Reviewer #2: Yes

5. Review Comments to the Author

Reviewer #1: The manuscript is well written though some editing is required in the grammar. The study is relevant because it established the variation and diversity in the application of markers of body adiposity and CMA in a Lebanese population. Some details is required in the methodology section. The authors should state clearly how they confirmed the nationality of the participants. Is it through any government instruments like National ID card, international passport etc. Examples of diseases affecting vitamin D metabolism should be stated and how individuals with these diseases were excluded.

Reviewer #2: This study has an interesting and relevant research question about the relevance of various measures of adiposity to cardiometabolic abnormalities in an understudied population. It is also important that the authors have attempted to identify the optimal cut-off points for these measures, as the established ones mainly come from European populations. The statistical analysis could benefit from major changes and the aim of the study needs to be placed more clearly in the context of related research in Middle East.

Major comments:

• Introduction: I think the introduction should include more information on cardiometabolic abnormalities and their trends in the Middle East area and what gaps of research there are for such populations. Are there other similar studies in such populations?

• Statistical analysis:

o Since this is a study about prediction, it would be important to follow relevant guidelines about prediction models i.e. TRIPOD (https://pubmed.ncbi.nlm.nih.gov/25560730/ ) or STARD both for conducting the analyses and reporting them.

o Lines 156-158: While discrimination is important to assess as a start, AUCs have also caveats (see for reference: https://www.ahajournals.org/doi/10.1161/circulationaha.106.672402 ), so prediction modelling should include also measures of calibration and other relevant e.g. reclassification. Please refer to the TRIPOD guidelines.

o Line 161: Considering that you have multiple predictors, thus multiple testing, consider performing a correction for multiple testing.

o The analyses presented in Table 3 are unadjusted. It might be more informative if the p-values come from analyses adjusted for relevant factors e.g. age, socioeconomic status, and smoking. You might want to consider presenting the mean differences and their 95% CI as well.

o When assessing prediction e.g. through discrimination (AUCs) it might be more clinically relevant to include some baseline predictors in the model e.g. age, education, SES and smoking and then assess discrimination with these predictors only and then also when adding the anthropometric markers. In this case you could also assess risk reclassification.

• Limitations paragraph:

o The authors mention “power analysis was fulfilled”. Could you include the power analysis in the methods section since it was done and especially since this is a relatively small study compared to other prospective studies nowadays?

o What about loss of predictive value from potential predictors not available in the study e.g. alcohol, diet and physical activity?

• Conclusions paragraph: Here it would be appropriate to add direction for future research including large prospective cohort studies and inclusion of more predictors.

Minor comments:

• Line 55: Change “low-density” to “high-density”

• Line 58: Do the cited references include the potential causes (especially stress exposure) of the rise in the metabolic syndrome prevalence? If not the part of the sentence “, is rising due to alterations in the lifestyle, dietary choices, and stress exposure” can be deleted or be supported by a reference which includes this.

• Line 60: References 7-8 are from approximately 15 years ago, while the sentence refers to global estimates of metabolic syndrome. Please use recent references and include the time period your estimates refer to for relevance.

• Line 76: Reference 18 seems irrelevant (Occurrence of resistance to antibiotics, metals, and plasmids in clinical strains of Staphylococcus spp)

• Lines 154-155: It is not clear which exposures were categorical, please specify.

• Line 155: t-test is a parametric method; were the assumptions met e.g. equal variances between the groups?

• Since you stratified your main analyses by sex, it might be sensible to do the same for Table 1 and Table 2.

• Table 2: Since mean and median are close, it would be enough to present only mean (SD) (also min, max not necessary) and you could actually include these in Table 1.

• Lines 178-179: “this was more common in males than females (47.5% vs. 59.8%)”, does 47.5% refer to females then? This is not clear; if this is the case please rephrase as follows: : “this was more common in males than females (59.8% vs 47.5% respectively)”

• Figures 1-3 could be combined in a three-panel figure.

6. PLOS authors have the option to publish the peer review history of their article (what does this mean?). If published, this will include your full peer review and any attached files.

**Do you want your identity to be public for this peer review?** For information about this choice, including consent withdrawal, please see our Privacy Policy.

Reviewer #1: No

Reviewer #2: **Yes: **Eirini Trichia

---

## [Decision Letter · Decision Letter 1]

20 Sep 2023

PGPH-D-23-00803R1

Abdominal volume index, waist-to-height ratio, and waist circumference are optimal predictors of cardiometabolic abnormalities in a sample of Lebanese adults: A cross-sectional study

Dear Dr. Abboud,

Thank you for submitting your manuscript to PLOS Global Public Health. After careful consideration, we feel that it has merit but does not fully meet PLOS Global Public Health’s publication criteria as it currently stands. Therefore, we invite you to submit a revised version of the manuscript that addresses the points raised during the review process.

We look forward to receiving your revised manuscript.

Kind regards,

Derrick Anthony Bennett

Academic Editor

Journal Requirements:

Additional Editor Comments (if provided):

The authors should address all of the comments regarding calibration and presentation of the results of the modelling. A more suitable correction for multiple testing, (if one is indeed needed) such as the Benjamini-Hochberg should be considered.

Reviewers' comments:

Reviewer's Responses to Questions

**Comments to the Author**

1. If the authors have adequately addressed your comments raised in a previous round of review and you feel that this manuscript is now acceptable for publication, you may indicate that here to bypass the “Comments to the Author” section, enter your conflict of interest statement in the “Confidential to Editor” section, and submit your "Accept" recommendation.

Reviewer #1: All comments have been addressed

Reviewer #2: (No Response)

2. Does this manuscript meet PLOS Global Public Health’s publication criteria? Is the manuscript technically sound, and do the data support the conclusions? The manuscript must describe methodologically and ethically rigorous research with conclusions that are appropriately drawn based on the data presented.

Reviewer #1: Yes

Reviewer #2: Partly

3. Has the statistical analysis been performed appropriately and rigorously?

Reviewer #1: Yes

Reviewer #2: No

4. Have the authors made all data underlying the findings in their manuscript fully available (please refer to the Data Availability Statement at the start of the manuscript PDF file)?

Reviewer #1: Yes

Reviewer #2: Yes

5. Is the manuscript presented in an intelligible fashion and written in standard English?

Reviewer #1: Yes

Reviewer #2: Yes

6. Review Comments to the Author

Reviewer #1: (No Response)

Reviewer #2: I would like to thank the authors for considering my comments and making an effort to address them. I have included some further suggestions.

It is good the authors considered their prediction analyses more carefully accounting for the TRIPOD guidelines. Estimation of reclassification measures could be done to complement other measures, but it is not necessary, especially since they also have limitations as the authors indicated. However, it would be important to show calibration of the prediction models and it is recommended that the authors use calibration plots for that also as per the TRIPOD guidelines (observed vs predicted risk within deciles of the predicted risk).

Lines 66-73: The authors have now added more information on Middle East and Lebanon, but the references on the prevalences of metabolic syndrome in the Middle East and Lebanon seem to be old (~15 years ago) although the authors state “in recent years”. Is it possible to cite a more recent reference for the prevalences of metabolic syndrome to make the context more relevant? Also when a prevalence is stated it would be good to mention the year it refers to as well.

Statistical analysis

Lines 182-185 could be added to the next paragraph, as they are related to predictive modelling as well. The c-statistic and AUC are the same thing, so they shouldn’t be presented as something different. It would be good that the authors also state how they determined the optimal cut-offs.

Lines 191-192: From this sentence, I understand that the Bonferroni adjustment was performed, because the models were adjusted over multiple variables. That is why the authors divided the alpha by 8 since there are 8 covariates. However, correction for multiple testing does not have to do with the number of covariates adjusted in the models, but with the number of tests that are considered in a group of analyses i.e. the number of models used. For example, in the total sample, 11 tests are performed (11 outcomes) and the same by sex. Also Bonferroni correction might be a little too strict for this analysis and the authors could consider the Benjamini-Hochberg correction for the false-discovery rate instead (https://www.jstor.org/stable/2346101 ) and especially performed by groups/families of analyses (https://www.ncbi.nlm.nih.gov/pmc/articles/PMC9945647/ ).

Tables 3 and 4: The main point of Table 3 is to show whether any of the anthropometric indices any more information over sociodemographic and lifestyle predictors in the risk discrimination for CMA for most of them. For example, the highest difference in the c-statistic for men is 0.85-0.772=0.078, while for women it is 0.028. The authors could present c-statistics with their 95% CI and maybe C-statistic differences and their 95% CI between Model 1 and each of the rest of the models and then comment on which differences are statistically significant, but also whether they think that the differences are also clinically meaningful. Table 4 could be included in the supplement, as it does not add much information over what is presented in Table 3. The new paragraph added to the results on Table 3 focuses on the main predictors for CMA individually, but apart from that, it would be good to discuss how adding the anthropometric indices improves the discrimination and calibration on top of the predictors in model 1. This whole comparison seems relevant from a public health or a clinical point of view, as it is easier and cheaper to predict something using sociodemographic and lifestyle factors and any additional cost or effort to measure anthropometric indices should be justified by a substantial increase in the predictive ability of the model.

Figure 1: These ROC curves could include the one that corresponds to Model 1 and then those from models when adding each anthropometric index to Model 1 (so same models as in Table 3). In this way the reader would also be able to visually inspect differences in risk discrimination by models.

Discussion: The discussion should be edited and framed around the updated results i.e. adjusted results, after correction for multiple testing and when it comes to prediction, how much information do anthropometric indices add on top of sociodemographic and lifestyle factors? Same for the abstract.

Line 65: Delete characters “o0”

7. PLOS authors have the option to publish the peer review history of their article (what does this mean?). If published, this will include your full peer review and any attached files.

**Do you want your identity to be public for this peer review?** For information about this choice, including consent withdrawal, please see our Privacy Policy.

Reviewer #1: No

Reviewer #2: No

---

## [Decision Letter · Decision Letter 2]

14 Nov 2023

PGPH-D-23-00803R2

Abdominal volume index, waist-to-height ratio, and waist circumference are optimal predictors of cardiometabolic abnormalities in a sample of Lebanese adults: A cross-sectional study

Dear Dr. Abboud,

Thank you for submitting your manuscript to PLOS Global Public Health. After careful consideration, we feel that it has merit but does not fully meet PLOS Global Public Health’s publication criteria as it currently stands. Therefore, we invite you to submit a revised version of the manuscript that addresses the points raised during the review process.

We look forward to receiving your revised manuscript.

Kind regards,

Derrick Anthony Bennett

Academic Editor

Journal Requirements:

Additional Editor Comments (if provided):

Please can you address all of the comments from the reviewer - particularly in respect to using the appropriate wording in the results and the discussion sections.

Reviewers' comments:

Reviewer's Responses to Questions

**Comments to the Author**

1. If the authors have adequately addressed your comments raised in a previous round of review and you feel that this manuscript is now acceptable for publication, you may indicate that here to bypass the “Comments to the Author” section, enter your conflict of interest statement in the “Confidential to Editor” section, and submit your "Accept" recommendation.

Reviewer #2: (No Response)

2. Does this manuscript meet PLOS Global Public Health’s publication criteria? Is the manuscript technically sound, and do the data support the conclusions? The manuscript must describe methodologically and ethically rigorous research with conclusions that are appropriately drawn based on the data presented.

Reviewer #2: Partly

3. Has the statistical analysis been performed appropriately and rigorously?

Reviewer #2: Yes

4. Have the authors made all data underlying the findings in their manuscript fully available (please refer to the Data Availability Statement at the start of the manuscript PDF file)?

Reviewer #2: Yes

5. Is the manuscript presented in an intelligible fashion and written in standard English?

Reviewer #2: Yes

6. Review Comments to the Author

Reviewer #2: I would like to thank the authors once again for addressing my points in detail. In my opinion the manuscript has substantially improved, but I still have a few comments for consideration.

Abstract-Methods: Change “c-statistic and 95% confidence intervals (CIs)” to “c-statistic [95% confidence intervals (CIs)] and calibration plots”

Abstract-Results: I do not agree with the statement “Adding anthropometric indices to sociodemographic variables improved model discrimination”. From table 3, we can see that the CIs of the c-statistic of the model with sociodemographic variables alone overlap with the CIs of the c-statistic of the model that additionally includes any of the anthropometric indices, which means that the two c-statistics do not significantly differ (which we can also see from the ROC curves). The p-values you have included in the table refer to the overall contribution of the anthropometric indices to the model rather than their incremental value over sociodemographic and lifestyle factors. I suggest instead to rewrite the statement as follows: “Adding anthropometric indices to sociodemographic variables did not significantly improve the model discrimination”.

Introduction – 3rd paragraph: Thank you for including more recent references. You can consider including information from this paper https://www.sciencedirect.com/science/article/pii/S0168822722007380 , which is from 2022 and highlights that “Eastern Mediterranean WHO regions had the highest prevalence across different MetS definitions”.

Statistical analysis:

Please rewrite the sentence “In order to compare… in two groups” as “The Chi-square and Fisher exact tests were used to assess associations between categorical variables and clinical metabolic abnormalities (CMA), and the Student t-test was used to assess associations between continuous variables and CMA.”

Thank you for providing an explanation on how you applied the FDR correction, which is helpful for the reviewers, but I think it is not necessary to be included in the manuscript. You can change the text “To correct for multiple testing…are not considered significant” to “To correct for multiple testing the Benjamini-Hochberg method was used for the analyses in the total sample and then in each sex separately (three groups of analyses with 11 tests performed in each group.” . Then in the end of the sentence you can add the appropriate reference i.e. “Benjamini Y, Hochberg Y (1995) Controlling the false discovery rate: a practical and powerful approach to multiple hypothesis testing. J R Stat Soc B 57:289–300”.

Please replace the “coefficients of the linear equation” with “calibration slope”.

Results – sample description: Insert “were” between “and” and “employed”.

Replace “31.6% smoke waterpipe” with “31.7% smoke waterpipe” (as per table 1 numbers).

Replace “stratifying between sex” with “stratifying by sex”.

Insert “for” after “except”.

Results – Table 3 text: When presenting results from Table 3, please delete the sentences “When adding the anthropometric indices the discrimination improved”, “When adding the anthropometric indices, the discrimination was improved from 0.86 (sociodemographic variables) to 0.89 (WC and WHtR).” and “When adding the anthropometric indices, the discrimination was improved from 0.77 (sociodemographic variables) to 0.85 (AVI and WC)”. Instead, please add the sentence “When adding the anthropometric indices to the model with the sociodemographic variables, the discrimination did not significantly improve in the total sample or analyses stratified by sex.” right before the sentence “The discrimination power … in Figure 1”. Please see my “Abstract-Results” comment for the rationale for this change.

Table 4: This table could go to the supplementary material. It is enough to have the calibration plots in the main set of figures.

Discussion – 1st paragraph: Please replace the sentence “Finally … sociodemographic variable alone” with the sentence “Finally, adding anthropometric indices to models with sociodemographic variables did not significantly improve the discriminatory ability of the model”.

Discussion – 6th paragraph: Please add “,so the results, especially the reported prevalences and cut-off points, might not be generalizable to the general Lebanese population” after “relative small samples”. I also think that you can delete the sentence “Finally, although… for risk prediction”, because you complemented the discrimination analysis with the calibration, so you do not need to list it as a limitation.

Figure 1: Panel A – Add “Overall” below “ROC curve”. Panels B and C – Remove “Gender:”. All panels- Replace “Source of the curve” with “Predictors included”. Even better, you could remove the legends from panels A and B and keep it only in Panel C and put all three plots in the same row. Apart from the sociodemographic variables it might be helpful to write the other legends as “+Body fat percent”, “+Conicity index” etc. It might be better to remove the grid from the figure background, so that the figures are cleaner.

Figure 2 (and same for Figures S1 and S2): On the top of each plot, keep only the variable name without underscore and “observed” and include the unit in parentheses unless it’s an index or a ratio e.g. “Body fat (%)”. As suggested for Figure 1, please remove the grids to have cleaner figures.

7. PLOS authors have the option to publish the peer review history of their article (what does this mean?). If published, this will include your full peer review and any attached files.

**Do you want your identity to be public for this peer review?** For information about this choice, including consent withdrawal, please see our Privacy Policy.

Reviewer #2: **Yes: **Eirini Trichia

---

## [Editor Report · Decision Letter 3]

28 Nov 2023

Abdominal volume index, waist-to-height ratio, and waist circumference are optimal predictors of cardiometabolic abnormalities in a sample of Lebanese adults: A cross-sectional study

PGPH-D-23-00803R3

Dear Dr Abboud,

We are pleased to inform you that your manuscript 'Abdominal volume index, waist-to-height ratio, and waist circumference are optimal predictors of cardiometabolic abnormalities in a sample of Lebanese adults: A cross-sectional study' has been provisionally accepted for publication in PLOS Global Public Health.

Best regards,

Derrick Anthony Bennett

Academic Editor